# Quantitative Predictive Studies of Multiple Biological Activities of TRPV1 Modulators

**DOI:** 10.3390/molecules29020295

**Published:** 2024-01-05

**Authors:** Xinmiao Wei, Tengxin Huang, Zhijiang Yang, Li Pan, Liangliang Wang, Junjie Ding

**Affiliations:** 1State Key Laboratory of NBC Protection for Civilian, Beijing 102205, China; weixm1999@163.com (X.W.); h1064482785@163.com (T.H.); yzjkid9@gmail.com (Z.Y.); bk6180b@163.com (L.P.); 2School of Physics and Electronic Engineering, Sichuan University of Science & Engineering, Zigong 643000, China

**Keywords:** machine learning, QSAR, TRPV1 channel, TRPV1 regulators, activity prediction

## Abstract

TRPV1 channel agonists and antagonists, which have powerful analgesic effects without the addictive qualities associated with traditional analgesics, have become a focus area for the development of novel analgesics. In this study, quantitative structure–activity relationship (QSAR) models for three bioactive endpoints (K_i_, IC_50_, and EC_50_) were successfully constructed using four machine learning algorithms: SVM, Bagging, GBDT, and XGBoost. These models were based on 2922 TRPV1 modulators and incorporated four types of molecular descriptors: Daylight, E-state, ECFP4, and MACCS. After the rigorous five-fold cross-validation and external test set validation, the optimal models for the three endpoints were obtained. For the K_i_ endpoint, the Bagging-ECFP4 model had a *Q*^2^ value of 0.778 and an *R*^2^ value of 0.780. For the IC_50_ endpoint, the XGBoost-ECFP4 model had a *Q*^2^ value of 0.806 and an *R*^2^ value of 0.784. For the EC_50_ endpoint, the SVM-Daylight model had a *Q*^2^ value of 0.784 and an *R*^2^ value of 0.809. These results demonstrate that the constructed models exhibit good predictive performance. In addition, based on the model feature importance analysis, the influence between substructure and biological activity was also explored, which can provide important theoretical guidance for the efficient virtual screening and structural optimization of novel TRPV1 analgesics. And subsequent studies on novel TRPV1 modulators will be based on the feature substructures of the three endpoints.

## 1. Introduction

TRPV1 channels are nociceptors found on C and Aδ fibers [1]. They detect various noxious stimuli, such as high temperatures (>42 °C), acidity (H^+^), and a range of endogenous and exogenous ligands [2]. These channels are crucial in pain management. TRPV1 modulators, including agonists and antagonists, have demonstrated significant efficacy in the treatment of neuropathic pain, osteoarthritis, and cancer pain [3]. Among them, TRPV1 agonists produce long-lasting and reversible analgesia through calcium-dependent desensitization, rendering TRPV1-expressing nerve fibers unresponsive to noxious stimuli [4]. TRPV1 antagonists, on the other hand, reduce nociceptive hypersensitivity by inhibiting TRPV1 channels, thus inhibiting the production of noxious sensations. Traditional analgesics, such as opioid narcotic analgesics and nonsteroidal anti-inflammatory analgesics, while initially providing temporary or partial pain relief, are associated with dose-limiting side-effects, lack of tolerance, and decreased efficacy over time, particularly impacting the treatment of chronic pain in the elderly [5]. However, existing TRPV1 modulators have side effects like strong irritation and hyperthermia, limiting their long-term clinical application [6,7]. Therefore, it is still necessary to develop novel TRPV1 modulators.

The agonistic or inhibitory activity of TRPV1 modulators is generally quantified using the concentration for 50% of maximal effect (EC_50_) or half maximal inhibitory concentration (IC_50_). K_i_ is the inhibition constant, which is a more precise indicator than IC_50_. The experimental approach to detect the effect of TRPV1 regulators on the opening (agonism) or closing (antagonism) of TRPV1 channels commonly involves the use of FLIPR [8] and electrophysiological membrane clamp [9]. Although intuitively clear, these experimental methods require the synthesis of the compound to be tested first and later assayed on TRPV1-expressing cells. Moreover, the speed of drug discovery is limited by the experimental methods, although high-throughput screening and combinatorial chemistry have been developed. Both the in vitro experiments, especially electrophysiological assays, require significant time and high investment costs.

The discovery of hits is a mandatory pathway to the discovery of novel TRPV1 modulators. However, high-throughput screening based on wet assays, combinatorial chemistry, and fragment-based drug design requires significant labor, material, and time costs and consumes a lot of effort on inactive compounds [10,11]. In recent years, computer-aided drug design, represented by quantitative structure–activity relationships (QSARs), has been rapidly developed due to the rise of artificial intelligence and big data [11]. QSAR modeling is a mathematical or statistical methodology that establishes a quantitative mapping between molecular structure and biological activity that can be used to predict the biological activity of new compounds on specific targets [11]. This methodology has been widely used in the discovery of various drug hits. There have been some structural modification studies of TRPV1 regulators based on 3D-QSAR models. For instance, Kristam et al. [12] constructed a 3D-QSAR model using 62 piperazine-aryl derived TRPV1 compounds with good predictive performance (*Q*^2^ = 0.9, *R*^2^ = 0.75). Then, they used a Topomer-CoMFA method to construct a new 3D-QSAR model [13]. However, the predictive performance of the new model did not improve. Similarly, Wang et al. [14] constructed a 3D-QSAR model with good predictive performance (*Q*^2^ = 0.522, *R*^2^ = 0.839) using the CoMSIA method based on 236 TRPV1 antagonists. Although these 3D-QSAR models showed promising results, they require the superposition of molecular 3D conformations. Unfortunately, the effect of conformation overlap of 3D-QSAR method could seriously affect the robustness of the models. Furthermore, 3D-QSAR models are often limited to predicting the properties of compounds with similar structures, thus having poor generalization ability [15].

To address the above problems and build a model with good generalization ability and stability, this study successfully constructed several QSAR models based on multiple machine learning algorithms for the three activity endpoints (EC_50_ of TRPV1 agonists, IC_50_ of TRPV1 antagonists, and K_i_). The internal and external validation showed that the models have good predictive performance and generalization ability, which can provide high-quality virtual screening models for the development of novel TRPV1 modulators.

## 2. Results and Discussion

### 2.1. Chemical Space and Scaffold Analysis

To construct the QSAR model, it is important for the dataset to encompass a wide range of activity. The K_i_ dataset ranges from 5.76 to 10.00 in terms of pK_i_ values, the EC_50_ dataset ranges from 3.95 to 8.72 in terms of pEC_50_ values, and the IC_50_ dataset ranges from 4.04 to 9.40 in terms of pIC_50_ values. Therefore, the datasets for the three activity endpoints cover a broad span of activity, ranging from µM to nM. The activity distributions of the training and testing sets for the three activity endpoints, as indicated by the histograms (Figure 1A–C), closely resemble those of the total dataset. This suggests that the division of the dataset is reasonable with respect to activity distribution. In addition, principal component analysis (PCA) was utilized to represent the scaffold distribution of the compounds in both the training and test sets (Figure 1). Notably, the compound scaffolds representing the test set of the three endpoints were mainly distributed within the compound scaffolds of their corresponding training sets, and no more outliers appeared. Hence, the scaffold division of the test set proved suitable for evaluating the predictive performance and generalization capability of the QSAR model.

Table 1 lists the top ten carbon scaffolds with the highest numbers, the vast majority of which include an isobutane carbon scaffold structure corresponding to the neck group of the TRPV1 modulator, typically an amide, ureido, or thiourea, among others. The analysis of compound structures in the datasets showed 77 carbon scaffolds in the K_i_ dataset, 275 in the IC_50_ dataset, and 97 in the EC_50_ dataset. This indicates a significant diversity in structural composition across the datasets for the three endpoints. In contrast, the head and tail in the backbone are mostly cyclic structures, corresponding to the tail moiety that forms a hydrophobic interaction in TRPV1 modulators and the head moiety that mostly contains an aromatic ring. It is noteworthy that some special carbon scaffolds appear in the scaffold of EC_50_. First, the cyclohexane carbon scaffold, ranked in terms of content, is quite different from the generic structure of TRPV1 modulators, and can be designed as a head group to provide ideas for fragment-based drug design. Secondly, the carbon scaffolds ranked fifth and sixth in terms of content both appear to have a bridging ring structure and can be designed as a tail moiety that can provide strong van der Waals interactions and help increase binding affinity [16].

### 2.2. Feature Selection

To enhance the interpretability and accuracy of the model while minimizing training costs, a method known as recursive feature elimination based on random forest (RFE-RF) is employed for feature selection. Initially, RFE-RF utilizes the complete set of features from a descriptor or molecular fingerprint for modeling. Subsequently, it proceeds to eliminate the least significant feature iteratively, employing the remaining features for subsequent modeling steps. Finally, it selects the combination of features with the lowest *RMSE*_CV_.

As shown in Figure 2, the performance of the model gradually improves as the number of features increases, eventually reaching a plateau. The number of features selected varies across the three active endpoints for different descriptors or molecular fingerprints. The red dots identified by orange dashed lines in each subplot of Figure 2 indicate the selected feature combinations. In K_i_, the optimal number of features for Daylight, E-state, ECFP4, and MACCS was 53 (2.6% of original features), 30 (27.3% of original features), 82 (8.0% of original features), and 50 (30.1% of original features), respectively; in IC_50_, the optimal number of features for Daylight, E-state, ECFP4, and MACCS was 183 (8.9% of original features), 33 (30.0% of original features), 293 (28.6% of original features), and 68 (41.0% of original features), respectively; and in EC_50_, Daylight, E-state, ECFP4, and MACCS have the optimal number of features of 168 (8.2% of original features), 25 (22.7% of original features), 55 (5.4% of original features), and 22 (13.3% of original features), respectively. These 12 sets of features will be used as independent variables to construct 48 active prediction models using four machine learning algorithms, i.e., 16 models for each endpoint.

### 2.3. Evaluation of K_i_ Activity Prediction Models

The evaluation results of the 16 K_i_ activity prediction models are presented in Table 2. It can be observed that the internal validation results of different algorithms under the same descriptor are similar. Furthermore, there is a consistent trend in the internal validation results of different descriptors under the same algorithm, with ECFP4 showing the highest performance, followed by Daylight, MACCS, and E-state. The model constructed using the Bagging algorithm and ECFP4 descriptors demonstrates the highest performance (*Q*^2^ = 0.778, *R*^2^ = 0.780), while the models constructed using SVM and E-state descriptors exhibit the lowest performance (*Q*^2^ = 0.502, *R*^2^ = 0.536). The *MAE*_CV_ and *MAE*_T_ of the vast majority of the models were in the range of 0.3–0.4, indicating that the difference between the predicted results and the experimental values was not more than half an order of magnitude. Thus, the predicted values of these models are of practical significance. Subsequently, the external validation results of the 16 models align with the internal validation results, reaffirming their good generalization ability and reliable prediction capability for the K_i_ activity values of new chemical entities. Figure 3 displays the scatter plot of the predicted values of the optimal model against the experimental values. The green dots represent the training set, while the orange dots represent the test set.

### 2.4. Evaluation of IC_50_ Activity Prediction Models

Table 3 presents the evaluation results of 16 IC_50_ activity prediction models. Among SVM, Bagging, and XGBoost, the prediction performance of the four descriptors is ranked as follows: ECFP4 > Daylight > MACCS > E-state. However, in GBDT, the prediction performance of Daylight is stronger than ECFP4. Comparing the internal validation results of K_i_ with those of the four algorithms in the IC_50_ dataset, there are significant differences. Specifically, the internal validation performance of GBDT is significantly lower than that of the other three algorithms. It is worth noting that, although the SVM model using E-state descriptors has the worst internal validation performance (*Q*^2^ = 0.487 ± 0.008, *RMSE*_CV_ = 0.424 ± 0.005, and *MAE*_CV_ = 0.338 ± 0.004) among all the models, the model constructed by XGBoost and ECFP4 is considered the optimal model for IC_50_ activity prediction. This model performs the best for both internal and external validation, demonstrating good generalization performance. Furthermore, the *MAE* of the IC_50_ model is comparable to that of the K_i_ prediction model, with errors within half an order of magnitude. Figure 4 illustrates the scatterplot of predicted versus experimental values for the optimal model, with green dots indicating the training set and orange dots representing the test set. The green solid line represents the trend line for the training set, while the orange solid line corresponds to the trend line for the test set. Notably, the trend lines of the training and test sets resemble those of the K_i_ prediction model, further confirming the model’s strong generalization ability.

### 2.5. Evaluation of EC_50_ Activity Prediction Models

Table 4 presents the performance evaluation results for the 16 EC_50_ activity prediction models. It is evident that the internal validation of the same descriptors varies less across different algorithms. However, the performance of the E-state descriptor in the SVM model is noticeably inferior to the other three algorithms. Additionally, the performance of the four descriptors in the Bagging, GBDT, and XGBoost algorithms follows the order of ECFP4 > Daylight > MACCS > E-state. Conversely, in the SVM algorithm, the internal validation of Daylight outperforms that of ECFP4, making it the optimal model among the EC_50_ activity prediction models with an external validation *R*^2^ exceeding 0.8, thus highlighting its exceptional predictive capability. Figure 5 illustrates the scatter plots comparing the predicted and experimental values for the SVM and Daylight models.

The results of internal and external validation for our models demonstrate a significantly reduced difference (less than 0.03) between *Q*^2^ and *R*^2^ compared to the previous 3D-QSAR model [12,13,14] (Table 5), which exhibited a difference of more than 0.25. This indicates a significant improvement in the generalization ability of the model. The enhanced performance can be attributed, primarily, to the larger dataset utilized in this study, as well as the robust stability of the machine learning algorithms employed. Notably, the model proposed by Kristam was developed using only 62 molecules, making it challenging to ensure generalizability.

### 2.6. Y-Randomization Test

Feature selection involves selecting the best performing feature combinations from high-dimensional descriptors and molecular fingerprints to build models that are highly fitted to experimental values. However, it is possible to obtain such models by chance, without any real correlation between the descriptors and experimental values. To assess the chance correlation of the model, we applied the Y-randomization test. During the Y-randomization test, the experimental values of pK_i_, pIC_50_, and pEC_50_ are randomly disrupted, destroying the original relationship between the descriptors or molecular fingerprints and the activity values, but the distribution of the activity values does not change [17]. We then re-modeled the disrupted data using the algorithm of the three optimal models and the molecular fingerprints, repeating the process 1000 times. The results of evaluating the 1000 randomized models using *Q*^2^ are shown in Figure 6. In this figure, the horizontal coordinate represents *Q*^2^, and the vertical coordinate represents the number of frequencies. The green bars on the left side of the three subfigures represent the histograms of the distribution of *Q*^2^ for the 1000 randomized models, while the orange vertical lines indicate the *Q*^2^ of the original models. From Figure 6, it is evident that all the *Q*^2^ of the randomized models fall between −1 and 0, indicating no correlation between the true value of the randomized model and the descriptor or molecular fingerprints. According to the paired-sample t-test, the confidence level of the randomized model compared to the original model is 99% (*p* < 0.001), which is statistically significant. Therefore, in the three optimal activity prediction models constructed in this paper, there exists a real correlation, rather than a chance correlation, between the modeled molecular fingerprints and K_i_, IC_50_, or EC_50_.

### 2.7. Model Interpretation

In this paper, we aim to interpret the three optimal models by ranking the importance of features. To select the features, we employ the RFE-RF method, which calculates the Gini index for each feature to indicate its significance. Figure 7 displays the five features with the highest importance among the three optimal models. Additionally, it is worth mentioning that the optimal models of K_i_ and IC_50_ utilize the ECFP4 fingerprint, whereas the optimal model of EC_50_ utilizes the Daylight fingerprint. Notably, the sum of the importance of all the features in the models is equal to 1.

The first five features of K_i_ had a cumulative importance of 0.493. Out of the dataset, 349 compounds had these features in the following descending order: 200, 667, 573, 316, and 997. This accounted for 52.80% of the total K_i_ dataset. Figure 8A displays the histogram of pK_i_ distribution for compounds containing the top 5 features. It is evident that the pK_i_ of these compounds is shifted one unit to the right compared to other compounds. The structure of the first five features is shown in Figure 9A. The structure of TRPV1 modulators typically comprises three parts: the head, the neck, and the tail. Generally, the head serves as a hydrogen bond acceptor, while the neck acts as a hydrogen bond acceptor and is commonly an amide, urea, or thiourea. On the other hand, the tail is a hydrophobic group [18,19]. In the case of these compounds, the first five characteristics correspond to the head (positions 667, 316, 997) and neck (positions 200, 573), with the head being a methylsulfonamide attached to a benzene ring and the neck being an amide group. Referring to the activity distribution in Figure 8A, it is reasonable to assume that compounds containing such a structure tend to exhibit high K_i_ activity and hold potential for modification.

The first three features of IC_50_ have been found to be significantly more important than the last two, with the order of importance being 672 bits > 128 bits > 378 bits. Therefore, the first three features are selected for model interpretation, as illustrated in Figure 7. In the IC_50_ dataset, there were 273 compounds (14.41% of the dataset) that demonstrated the first three features. The cumulative importance of these features amounted to 0.225. Figure 8B highlights that compounds possessing the first three features exhibited a rightward shift in pIC_50_ compared to other compounds, suggesting a higher level of antagonistic activity. As illustrated in Figure 9B, positional markers 672 and 128 correspond to the aromatic ring of the head and the urea group of the neck, respectively, while position 378 signifies the indole ring of the head. The indole in the head acts as both a hydrogen bond donor and acceptor, facilitating specific interactions for antagonist binding to TRPV1 channels. Therefore, compounds that incorporate 1*H*-indole in the head may potentially possess highly active antagonistic properties.

The Daylight fingerprint is different from ECFP4 in that it represents the molecular structure as a linear path from atoms, and thus has no central atom. The importance of the 67-position feature is 0.311, which is much higher than that of the other features, and the number of compounds with this feature is 205, which accounts for 55.86% of the EC_50_ dataset, thus this feature is used to interpret the model. In Figure 8C, the compounds with position 67 have a significant rightward shift in pEC_50_ compared to the other compounds, and the IC_50_ activity is nearly two orders of magnitude different. As can be seen in Figure 9C, the 67-position feature indicates the ureido group in the neck and the benzene ring in the head. This indicates that most of the highly active TRPV1 agonists have phenylurea at the neck and head, and thus compounds containing phenylurea are potentially highly active TRPV1 agonists.

## 3. Materials and Methods

### 3.1. Data Collection and Processing

Data on the K_i_, IC_50_, and EC_50_ activities of human TRPV1 channels were collected from the ChEMBL [20] and PubChem [21] databases. The data were processed according to the following steps: 1. compounds without a clear type of activity and activity value were removed; 2. the units of nanomoles (nM) or micromoles (μM) in the original data were converted to M and the negative logarithms with a base of 10 were taken (i.e., pK_i_, pIC_50_, and pEC_50_); 3. the compounds with multiple activity values were de-weighted, following the rule that if the maximum difference of the negative logarithm is less than or equal to 1, the mean value is taken as the activity value of the compound, and the compound is discarded otherwise; 4. salt ions and metal ions were removed from the dataset. After processing, three activity datasets were obtained, consisting of 661 K_i_, 1894 IC_50_, and 367 EC_50_ values. Based on these datasets, three QSAR models were constructed.

### 3.2. Descriptor Generation

This paper utilizes four different methods to extract features and build QSAR models: Daylight fingerprints, molecular access system (MACCS) [22] fingerprints, electrotopological state indices (E-state) molecular descriptors [23], and Extended-Connectivity Fingerprints (ECFPs) [24]. MACCS fingerprints are substructure-based molecular fingerprints, and this paper selects the commonly used 166-bit fingerprints. Daylight fingerprints, also known as path-based molecular fingerprints, characterize molecules through different atomic paths represented by a total of 2048 bits. E-state molecular descriptors simultaneously characterize the molecular structure and electrical characteristics with a total of 110 features. ECFP fingerprint is a circular topological fingerprint based on Morgan’s algorithm. This study uses ECFP4 with a diameter of 4 and 1024 bits. These descriptors of compounds in databases were calculated through the Scopy [25] and rdkit [26] toolkit.

### 3.3. Data Set Segmentation

In order to avoid training bias or overfitting and to maintain similar structural distribution of compounds in each subset close to each other, this paper divides the dataset into training and test sets according to the carbon scaffold. The carbon scaffold is determined by removing all R groups from the molecule and retaining only the connecting groups between the ring systems, while converting heteroatoms to carbon atoms and bonding sequences to single bonds. The Scopy toolkit [25] is employed to calculate the carbon scaffold of the compounds in this study. If there are less than five molecules with the same carbon scaffold, one molecule is randomly assigned to the training set and the remaining molecules are assigned to the test set. On the other hand, if there are five or more molecules with the same carbon scaffold, 80% of them are randomly assigned to the training set while the remaining 20% are assigned to the test set.

### 3.4. Machine Learning Methods

In this study, four machine learning algorithms are used for the construction of QSAR models, namely support vector machine (SVM), gradient boosting decision tree (GBDT), extreme gradient boosting (XGBoost) and bagging. The models of the 4 algorithms were implemented via the scikit-learn toolkit [27].

SVM [28] is a statistical learning algorithm based on the principle of Vapnik structural risk minimization. Originally developed for classification problems, SVM can also be extended to regression tasks by introducing slack variables. In the regression task, the objective is to find a hyperplane with a small number of paradigms, while minimizing the sum of the distances from the data to the hyperplane [29]. Its high degree of generalization ability has contributed to its increasing popularity in the QSAR/QSPR species.

GBDT [30] is a machine learning algorithm based on the idea of Boosting integration. GBDT updates the strong learner by decreasing the loss function, fitting the loss approximation for each round of iteration with the negative gradient of the loss function. A disadvantage of GBDT is that it is difficult to train in parallel and is less efficient.

XGBoost was developed by Tianqi Chen et al. [31]. Other Boosting algorithms develop their models in a sequential phase manner like other Boosting algorithms. However, XGBoost enables parallel computation and also has improved handling of missing values compared to GBDT. In addition, XGBoost is highly resistant to overfitting due to the inclusion of regular terms.

Unlike Boosting, each base learner in Bagging [32] is independent and can be computed in parallel. Bagging samples n sample sets using an autonomous sampling method and training a base learner for each sample set. Afterwards, the learners are combined. Hence, approximately 36.8% of the samples in the initial dataset do not appear in the sampling set. These samples can be used as a validation set to test the training performance and generalization ability of the model. The Bagging algorithm focuses on the reduction in variance and is known for its integration and efficiency.

The grid search in scikit-learn was used for parameter tuning. The key parameters of SVM are C (the penalty coefficient) and gamma (the coefficient of the kernel function). In grid search, the values of C were set as 0.01, 0.1, 1, 10, 100, and 1000; the values of gamma were set as 0.0001, 0.001, and 0.01; and the kernel was chosen as RBF. The number of decision trees is the parameter of Bagging, XGBoost, and GBDT ranging from 100 to 1000, with a step size of 50.

### 3.5. Performance Evaluation Indicators

To ensure the good generalization ability of the QSAR model in predicting the biological activity of new chemical entities, internal validation and external validation were conducted. The model was internally validated using five-fold cross-validation (CV) and independent test sets. In five-fold CV, the training set was divided into five equal parts, with four parts used for constructing the model and one part used for model validation. This process was repeated five times, allowing each part of the data to serve as a validation set. Four main statistical parameters were employed to evaluate the model’s performance: the coefficient of determination (*Q*^2^), the root mean square error (*RMSE*_CV_), and the mean absolute error (*MAE*_CV_) for CV and the coefficient of determination (*R*^2^), the root mean square error (*RMSE*_T_), and the mean absolute error (*MAE*_T_) for the test set. The formulas for *Q*^2^ (*R*^2^), *RMSE*, and *MAE* are given below:(1)$$Q^{2}\left(R^{2}\right)=1-\frac{\sum_{i=1}^{n}{\left({\hat{y}}_{i}-y_{i}\right)}^{2}}{\sum_{i=1}^{n}{\left({\bar{y}}_{i}-y_{i}\right)}^{2}}$$
(2)$$RMSE=\sqrt{\frac{1}{n}\sum_{i=1}^{n}{\left({\hat{y}}_{i}-y_{i}\right)}^{2}}$$
(3)$$MAE=\frac{1}{n}\sum_{i=1}^{n}\left|{\hat{y}}_{i}-y_{i}\right|$$
where ${\hat{y}}_{i}$ is the predicted value, $y_{i}$ is the true value, and ${\bar{y}}_{i}$ is the average value of $y_{i}$ in the sample. From the formula, it can be seen that the smaller the value of *RMSE* and *MAE*, the better the performance of the model; while the larger the value of *Q*^2^ or *R*^2^, the better the performance of the model.

## 4. Conclusions

To accelerate the discovery of novel TRPV1 modulators, QSAR models that can quantitatively predict K_i_, IC_50_, and EC_50_ were constructed using four machine learning algorithms based on 2922 biological activity data. After rigorous internal and external validation, the constructed models exhibited excellent external prediction performance and generalization ability. The model feature importance analysis revealed that the key feature structures of the three endpoints were concentrated in the head and neck of the molecule, aligning with the conclusion that the polar interactions between the TRPV1 regulator and TRPV1 only existed in their head and neck region. Specifically, a higher K_i_ activity tended to be observed in molecules with a methylsulfonamide attached to a benzene ring in the head and an amide group in the neck. Additionally, molecules containing 1*H*-indole in the head showed potential as highly active antagonists, while those containing phenylurea have a likely potential to be highly active TRPV1 agonists. These findings pertaining to the influence of the microstructure of TRPV1 modulators on their biological activities are expected to provide guidance for the rational design and efficient screening of novel analgesic drugs.

## Figures and Tables

**Figure 1 molecules-29-00295-f001:**
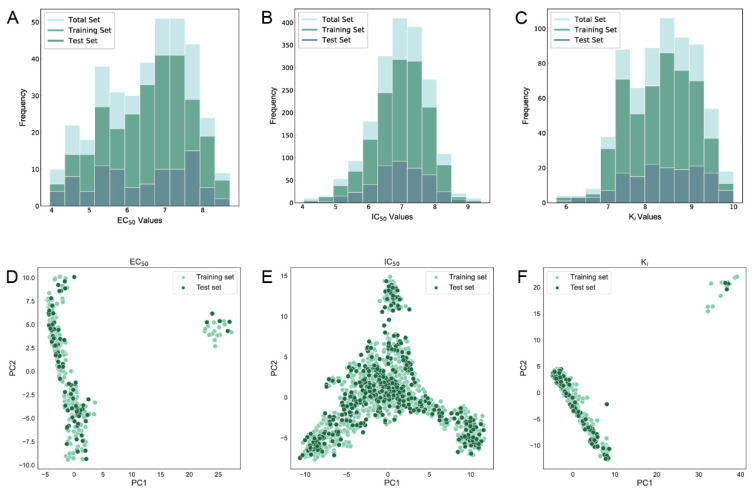
Distribution histograms (**A**–**C**) and principal component analysis plots (**D**–**F**) for the EC_50_, IC_50_, and K_i_ data sets.

**Figure 2 molecules-29-00295-f002:**
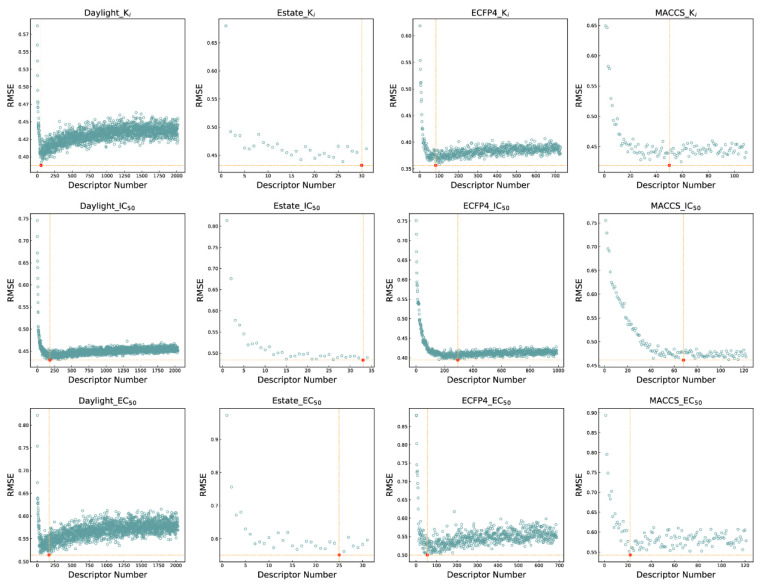
Feature selection results based on RFE-RF. The red dot indicates the point with the lowest *RMSE*, and the horizontal and vertical dotted lines refer to the *RMSE* and the number of features corresponding to this point respectively.

**Figure 3 molecules-29-00295-f003:**
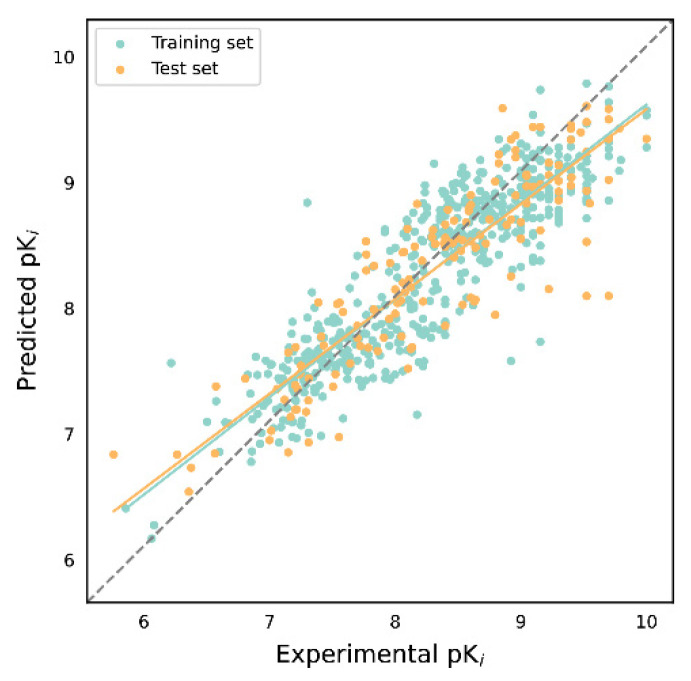
Scatter plot of predicted values versus experimental values of K_i_ prediction model based on Bagging and ECFP4. The green line indicates the trend line of the training set and the orange line indicates the trend line of the test set.

**Figure 4 molecules-29-00295-f004:**
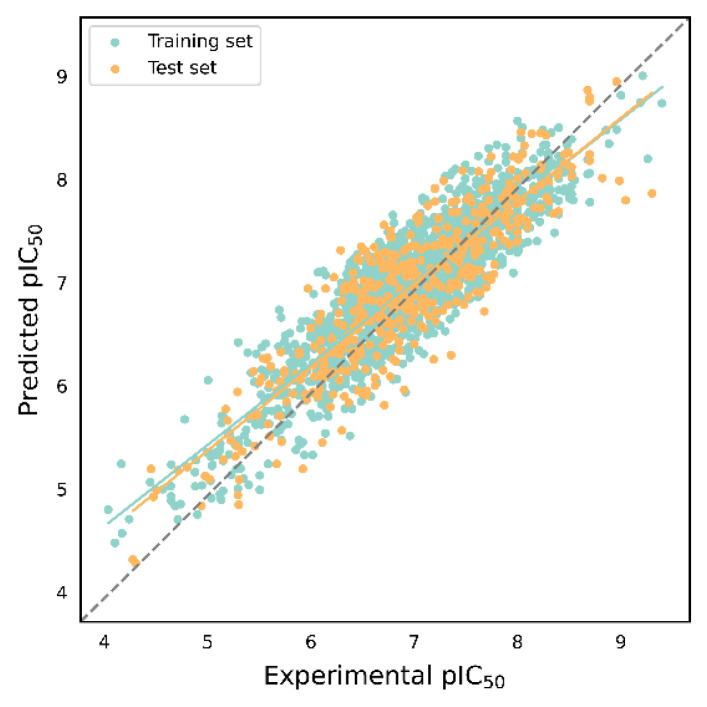
Scatter plot of predicted values versus experimental values of IC_50_ prediction model based on XGBoost and ECFP4. The green line indicates the trend line of the training set and the orange line indicates the trend line of the test set.

**Figure 5 molecules-29-00295-f005:**
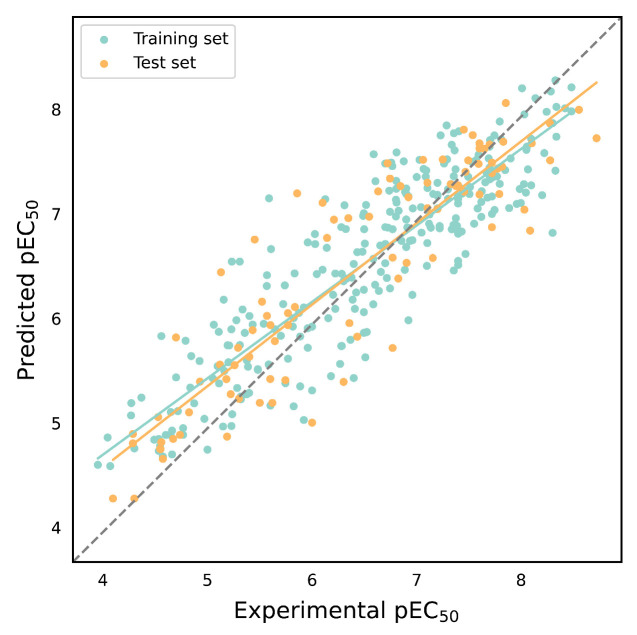
Scatter plot of predicted values versus experimental values of EC_50_ prediction model based on SVM and Daylight. The green line indicates the trend line of the training set and the orange line indicates the trend line of the test set.

**Figure 6 molecules-29-00295-f006:**
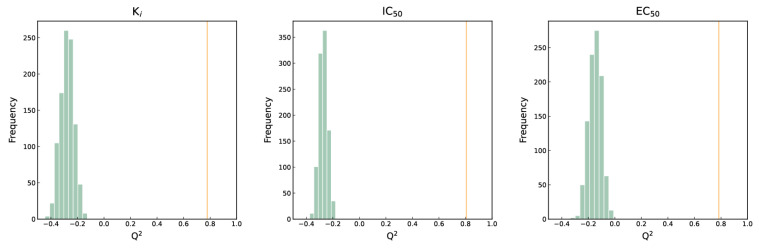
The distribution of *Q*^2^ of randomization models of three activity endpoints. the left-side green bar represents the randomized *Q*^2^ distribution, and the orange vertical line on the right side represents the *Q*^2^ of the original model.

**Figure 7 molecules-29-00295-f007:**
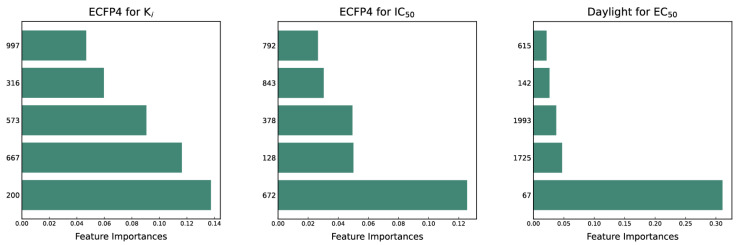
Descriptor importance of the top 5 features of 3 optimal models. The vertical coordinate represents the bit encoding of the molecular fingerprint, while the horizontal coordinate corresponds to the feature importance.

**Figure 8 molecules-29-00295-f008:**
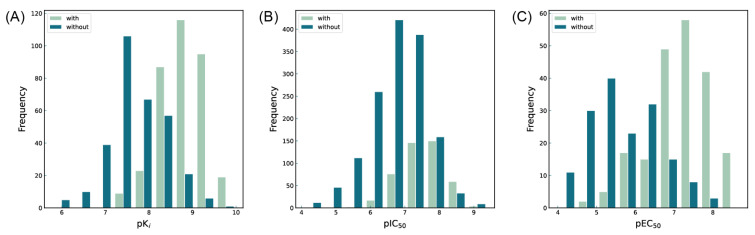
(**A**) Histogram of activity distribution of compounds with and without feature structures for pK_i_ endpoint; (**B**) Histogram of activity distribution of compounds with and without feature structures for pIC_50_ endpoint; (**C**) Histogram of activity distribution of compounds with and without feature structures for pEC_50_ endpoint.

**Figure 9 molecules-29-00295-f009:**
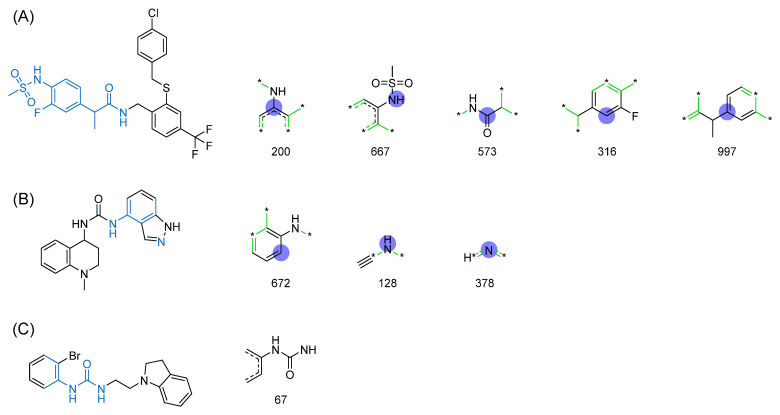
Feature structures and their position in the molecules. The blue structure on the left side of the molecule is the sum of the sub-structures corresponding to the features. On the right side, the featured structures are depicted, with the central atoms marked by purple dots. The atoms within the bonding radius are represented in black, while the green color is used to indicate the environments of the featured structures within the molecule. An asterisk indicates an unknown atom, which could be carbon, nitrogen, or something else. (**A**) The representative compound in K_i_ is shown on the left, and 5 substructures with the most importance are shown on the right. (**B**) The representative compound in IC_50_ and 3 substructures with the most importance. (**C**) The representative compound in EC_50_ and one substructure with the most importance.

**Table 1 molecules-29-00295-t001:** Top 10 carbon scaffolds and corresponding numbers of K_i_, IC_50_, and EC_50_ data sets.

No.	K_i_	IC_50_	EC_50_
Carbon Scaffold	Number	Carbon Scaffold	Number	Carbon Scaffold	Number
1	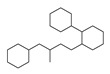	137	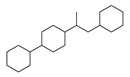	174	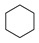	41
2	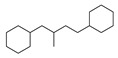	83	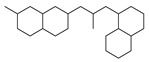	152	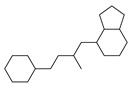	30
3	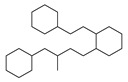	47	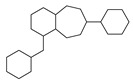	100	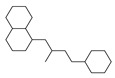	24
4	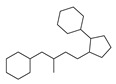	45	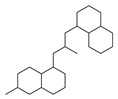	77	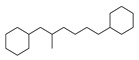	17
5	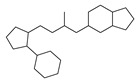	37	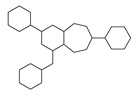	71	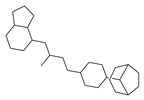	14
6	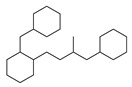	29	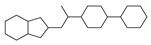	63	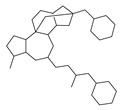	11
7	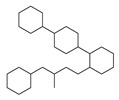	22	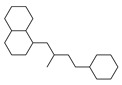	58	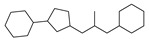	11
8	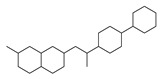	18	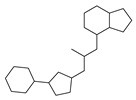	46	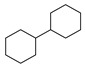	10
9	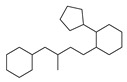	18	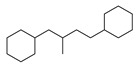	43	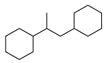	9
10	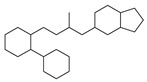	18	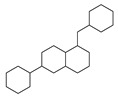	35	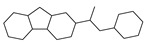	8

**Table 2 molecules-29-00295-t002:** The results of internal and external validation of K_i_ prediction models.

*Algorithm*	*Descriptor*	$Q^{2}$	${RMSE}_{CV}$	${MAE}_{CV}$	$R^{2}$	${RMSE}_{T}$	${MAE}_{T}$
SVM	Daylight	0.725 ± 0.012	0.408 ± 0.009	0.317 ± 0.005	0.766	0.419	0.320
	E-state	0.502 ± 0.010	0.550 ± 0.005	0.417 ± 0.006	0.536	0.590	0.448
	ECFP4	0.744 ± 0.008	0.394 ± 0.006	0.318 ± 0.004	0.761	0.424	0.325
	MACCS	0.684 ± 0.006	0.438 ± 0.004	0.344 ± 0.004	0.687	0.485	0.362
Bagging	Daylight	0.742 ± 0.018	0.395 ± 0.013	0.307 ± 0.009	0.779	0.408	0.312
	E-state	0.677 ± 0.018	0.442 ± 0.012	0.348 ± 0.008	0.642	0.519	0.393
	ECFP4	0.778 ± 0.012	0.367 ± 0.010	0.291 ± 0.008	0.780	0.407	0.305
	MACCS	0.697 ± 0.024	0.428 ± 0.016	0.334 ± 0.013	0.750	0.433	0.323
GBDT	Daylight	0.723 ± 0.010	0.410 ± 0.007	0.326 ± 0.005	0.755	0.429	0.332
	E-state	0.671 ± 0.013	0.447 ± 0.009	0.356 ± 0.007	0.623	0.532	0.410
	ECFP4	0.759 ± 0.007	0.382 ± 0.005	0.309 ± 0.004	0.757	0.427	0.329
	MACCS	0.686 ± 0.011	0.437 ± 0.008	0.340 ± 0.007	0.703	0.472	0.371
XGBoost	Daylight	0.723 ± 0.022	0.410 ± 0.015	0.317 ± 0.011	0.766	0.419	0.316
	E-state	0.683 ± 0.032	0.438 ± 0.020	0.342 ± 0.014	0.648	0.514	0.385
	ECFP4	0.771 ± 0.014	0.373 ± 0.011	0.301 ± 0.009	0.816	0.371	0.292
	MACCS	0.696 ± 0.020	0.429 ± 0.013	0.337 ± 0.011	0.745	0.437	0.330

**Table 3 molecules-29-00295-t003:** The results of internal and external validation of IC_50_ prediction models.

*Algorithm*	*Descriptor*	$Q^{2}$	${RMSE}_{CV}$	${MAE}_{CV}$	$R^{2}$	${RMSE}_{T}$	${MAE}_{T}$
SVM	Daylight	0.726 ± 0.006	0.424 ± 0.005	0.338 ± 0.004	0.744	0.443	0.353
	E-state	0.487 ± 0.008	0.580 ± 0.004	0.455 ± 0.003	0.545	0.590	0.455
	ECFP4	0.759 ± 0.006	0.398 ± 0.005	0.318 ± 0.004	0.763	0.426	0.342
	MACCS	0.639 ± 0.005	0.487 ± 0.004	0.381 ± 0.003	0.682	0.494	0.391
Bagging	Daylight	0.719 ± 0.016	0.429 ± 0.012	0.343 ± 0.008	0.712	0.469	0.366
	E-state	0.642 ± 0.020	0.485 ± 0.013	0.376 ± 0.010	0.628	0.534	0.426
	ECFP4	0.757 ± 0.015	0.399 ± 0.012	0.318 ± 0.008	0.722	0.462	0.362
	MACCS	0.674 ± 0.017	0.462 ± 0.011	0.364 ± 0.008	0.681	0.494	0.396
GBDT	Daylight	0.685 ± 0.007	0.455 ± 0.005	0.368 ± 0.003	0.706	0.475	0.378
	E-state	0.555 ± 0.006	0.540 ± 0.003	0.428 ± 0.002	0.584	0.564	0.449
	ECFP4	0.673 ± 0.004	0.463 ± 0.003	0.374 ± 0.003	0.703	0.477	0.386
	MACCS	0.579 ± 0.005	0.525 ± 0.003	0.418 ± 0.003	0.610	0.546	0.437
XGBoost	Daylight	0.742 ± 0.020	0.411 ± 0.015	0.325 ± 0.011	0.746	0.441	0.347
	E-state	0.660 ± 0.022	0.472 ± 0.014	0.368 ± 0.011	0.664	0.507	0.389
	ECFP4	0.806 ± 0.013	0.357 ± 0.011	0.290 ± 0.007	0.784	0.407	0.328
	MACCS	0.699 ± 0.020	0.444 ± 0.014	0.349 ± 0.009	0.727	0.457	0.367

**Table 4 molecules-29-00295-t004:** The results of internal and external validation of EC_50_ prediction models.

*Algorithm*	*Descriptor*	$Q^{2}$	${RMSE}_{CV}$	${MAE}_{CV}$	$R^{2}$	${RMSE}_{T}$	${MAE}_{T}$
SVM	Daylight	0.784 ± 0.009	0.505 ± 0.010	0.409 ± 0.008	0.809	0.532	0.420
	E-state	0.665 ± 0.013	0.629 ± 0.011	0.509 ± 0.012	0.716	0.649	0.492
	ECFP4	0.772 ± 0.008	0.518 ± 0.009	0.416 ± 0.006	0.844	0.481	0.382
	MACCS	0.758 ± 0.010	0.534 ± 0.011	0.423 ± 0.011	0.751	0.607	0.488
Bagging	Daylight	0.765 ± 0.015	0.527 ± 0.016	0.415 ± 0.013	0.718	0.647	0.492
	E-state	0.725 ± 0.022	0.570 ± 0.022	0.454 ± 0.015	0.735	0.626	0.474
	ECFP4	0.782 ± 0.017	0.507 ± 0.018	0.400 ± 0.016	0.844	0.480	0.367
	MACCS	0.746 ± 0.025	0.547 ± 0.025	0.431 ± 0.020	0.766	0.589	0.450
GBDT	Daylight	0.772 ± 0.014	0.518 ± 0.015	0.408 ± 0.013	0.745	0.614	0.465
	E-state	0.731 ± 0.019	0.563 ± 0.019	0.458 ± 0.012	0.777	0.575	0.428
	ECFP4	0.775 ± 0.012	0.515 ± 0.013	0.402 ± 0.011	0.832	0.499	0.404
	MACCS	0.742 ± 0.012	0.552 ± 0.012	0.432 ± 0.012	0.759	0.597	0.475
XGBoost	Daylight	0.771 ± 0.030	0.519 ± 0.030	0.409 ± 0.023	0.777	0.575	0.443
	E-state	0.729 ± 0.026	0.566 ± 0.025	0.439 ± 0.022	0.772	0.581	0.445
	ECFP4	0.778 ± 0.021	0.512 ± 0.022	0.395 ± 0.017	0.840	0.487	0.380
	MACCS	0.751 ± 0.019	0.542 ± 0.019	0.422 ± 0.016	0.699	0.668	0.501

**Table 5 molecules-29-00295-t005:** The results of internal and external validation of prediction models and previous studies.

	*K_i_ Model*	*IC_50_ Model*	*EC_50_ Model*	*Kristam et al.* *[12]*	*Wang et al.* *[14]*
$Q^{2}$	0.778	0.806	0.784	0.9	0.522
$R^{2}$	0.780	0.784	0.809	0.75	0.839
*n*	661	1894	367	62	236

## Data Availability

The data presented in this study are available in article.

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
