# Peer review of "Quantitative Predictive Studies of Multiple Biological Activities of TRPV1 Modulators"

_molecules, 2024, doi:10.3390/molecules29020295_

Round 1

Reviewer 1 Report

Comments and Suggestions for Authors

Summary:

This study focuses on developing quantitative structure-activity relationship (QSAR) models for TRPV1 channel agonists and antagonists, which offer potent analgesic effects without the addiction risks of traditional analgesics. Using 2922 TRPV1 modulators and four machine learning algorithms (SVM, Bagging, GBDT, XGBoost), the researchers successfully constructed models for three bioactive endpoints (Ki, IC50, and EC50) with high predictive performance and robustness. Additionally, the study explored the relationship between molecular substructures and biological activity, offering valuable insights for the virtual screening and structural optimization of new TRPV1 analgesics.

To justify my decision (Minor changes), I have listed approximately 15 comments, suggestions, and questions below. I hope these will assist the authors in enhancing the article for future submission. Please respond to these in a separate document.

1.       Abstract: Why do you call 'A five-fold cross validation ' the rigorous'? Also not sure if Q2 and R2 can show excellent robustness here.

2.       Line 65: the text "constructed a 3D-QSAR model " is repeated.

3.       Section 2.2. It is important to specify the software used in this study for generating descriptors. Please indicate whether a single software or multiple software tools were employed for this purpose. Regarding the feature extraction process, could you clarify whether these features are extracted from a larger database of features using specific variable selection methods, or if the descriptors are being generated independently? Additionally, the term 'Decorrelation' has not been adequately explained in this section. To enhance clarity, I recommend revising the subtitle from 'Feature Extraction and Decorrelation' to 'Descriptor Generation', or a similar term that more accurately reflects the content of the section.

4.       Section 2.3.: When talking about splitting molecules into test and model set. Is this choice of five molecules as a threshold supported or recommended by any references? Additionally, why was the number specifically chosen as five instead of six or four?

5.       Section 2.4. Again, could you please specify which software or suite of functions was utilized for the modeling? Additionally, could you provide more detailed information about the parameters used in training the models?

6.       Figure 1: Please explain what exactly the PCA score plots in Figure 1 represent. What does each dot signify? Does it represent a scaffold or a compound? Additionally, the PCA plots in Figures 1D and 1F are characterized by certain objects being far from the major group of molecules. Have you considered these objects as outliers? These objects could significantly influence the model.

7.       Table 1. The authors present the carbon scaffold in descending order, starting from the most abundant. However, have you tried to average (and report) the activity (Ki, IC50, EC50) for each group of cores? would be there any regularity after taking into consideration such approach?

8.       Lines 210-213: IIn addition to reporting the number of selected features, it would be beneficial to include their percentage values. For instance, if we originally have 100 features, selecting 50 features would constitute 50% of the total variables.

9.       Figure 2: Could you please specify what is plotted on the Y-axis in the graph? Is it the Root Mean Square Error (RMSE) for the model, test, or external test set? Or is it the cross-validation RMSE (RMSEcv)? This detail should be included in the figure caption.

10.   Table 2: How did you estimate the error when reporting Q2, RMSEcv, MAEcv?

11.   Tables 2-4: There are many numbers in those tables. I get the impression that readers who want to analyze/understand the data need another ML algorithm (or at least PCA) to see the trends in this data. Do you think we can do something about that?

12.   Line 283: Y-randomization is repeated phrase.

13.   Model Interpretation: This section lacks a comparison of the authors' results with the general knowledge available in the open literature regarding similar compounds or activities. It should also offer general guidelines on the characteristics of future, highly active molecules, providing direction for researchers involved in synthesizing such compounds. If the authors believe their discussion is sufficient, they should highlight this aspect more prominently in the text.

14.   RFE-RF: It would be beneficial to include more details about the Recursive Feature Elimination with Random Forest (RFE-RF) method in the Experimental section, as it appears to be a crucial approach utilized in this paper.

15.   Line 306: Please rewrite 'Gini importance of each feature to represent its importance' to 'Gini index for each feature to indicate its significance' (or importance).

16.   Line 322: what exactly you mean by '... featured WORD structures'. Please remove WORD.

17.   Lines 323-325: I recommend using a color other than gray for the bonds in Figure 9, as the gray bonds are only discernible upon zooming into the substructures.

18.   Figure 9. Please indicate in the caption a meaning of A, B, and C.

Author Response

Dear reviewer:

On behalf of my co-authors, we thank you very much for giving us an opportunity to revise our manuscript, and we also appreciate reviewers very much for their positive and constructive comments and suggestions on our manuscript entitled “Quantitative predictive studies of multiple biological activities of TRPV1 modulators”.

Thank you very much for your comments and professional advice. These opinions help to improve academic rigor of our article. Based on your suggestion and request, we have made corrected modifications on the revised manuscript. We hope that our work can be improved again. The responses to the review comments are as follows:

Q1: Abstract: Why do you call 'A five-fold cross validation ' the rigorous'? Also not sure if Q2 and R2 can show excellent robustness here.

Response: We consider 5-fold cross validation to be a rigorous validation of model performance because 5-fold cross validation divides the dataset into 5 parts and the average results of multiple 5-fold cross validations adequately represent the performance of the model. However, this statement is misleading as a qualifier of 5-fold cross-validation, we decided to delete "rigorous" (line 347), which does not affect our methodology and conclusions, but corrects the misleading statement. In addition, we rethought the meaning of robustness and clarified that robustness cannot be captured by R2 or Q2, so we removed the description of robustness (line 20).

Q2: Line 65: the text "constructed a 3D-QSAR model " is repeated.

Response: Thank you for the reviewer's reminder, it has been revised in the text, please see line 63 for details.

Q3: Section 2.2. It is important to specify the software used in this study for generating descriptors. Please indicate whether a single software or multiple software tools were employed for this purpose. Regarding the feature extraction process, could you clarify whether these features are extracted from a larger database of features using specific variable selection methods, or if the descriptors are being generated independently? Additionally, the term 'Decorrelation' has not been adequately explained in this section. To enhance clarity, I recommend revising the subtitle from 'Feature Extraction and Decorrelation' to 'Descriptor Generation', or a similar term that more accurately reflects the content of the section.

Response: Thank you for the reviewer's suggestion. The data preprocessing and four descriptors in the article were calculated using rdkit and Scopy, chemoinformatics toolkits written in Python, and which has been added to the article, please see line 310 in red. These features (descriptors) are calculated by rules for each molecule. Additionally, we have followed your suggestion and revised the subtitle from 'Feature Extraction and Decorrelation' to 'Descriptor Generation', please see line 300 in red.

Q4: Section 2.3.: When talking about splitting molecules into test and model set. Is this choice of five molecules as a threshold supported or recommended by any references? Additionally, why was the number specifically chosen as five instead of six or four?

Response: In fact, this is a common practice of partitioning datasets (ref. https://doi.org/10.1080/1062936x.2020.1765195, https://doi.org/10.1186/s13321-019-0383-2, https://doi.org/10.1016/j.jmgm.2013.08.014). The proportion of dividing the dataset should ensure the size of the test set. If the test set is too small, the predicted results of the test set will not be representative, while if the test set is too large, it will result in fewer data participating in model training, affecting prediction performance. As for 5 or 6, the difference is small and not significant, so choosing either one is acceptable.

Q5: Section 2.4. Again, could you please specify which software or suite of functions was utilized for the modeling? Additionally, could you provide more detailed information about the parameters used in training the models?

Response: Thanks for your valuable suggestions. (i)The program for building the model is written in Python language, where the machine learning algorithm calls the algorithm model integrated in scikit-learn. In addition, the recursive feature elimination based on random forest (RFE-RF) written in-house was used for feature selection.

(ii) The grid search was used for the tuning in this study, and the tuning process is supplemented in lines 347-351, which have been marked in red.

Q6: Figure 1: Please explain what exactly the PCA score plots in Figure 1 represent. What does each dot signify? Does it represent a scaffold or a compound? Additionally, the PCA plots in Figures 1D and 1F are characterized by certain objects being far from the major group of molecules. Have you considered these objects as outliers? These objects could significantly influence the model.

Response: The PCA plot is complementary to the histogram in Fig. 1, and is designed to test the consistency of the distribution of the data in the training set and the test set. Each point represents the position of a molecule in the two features PC1 and PC2 after PCA dimensionality reduction. From the results, the distributions of the training and test sets are consistent. In addition, certain objects far from the main group of molecules can be considered outliers, but if the molecules represented by these objects are removed, the model will not be able to accurately predict the activity of these molecules. The presence of outliers may also be due to the small amount of data on active compounds of TRPV1 channel.

Q7: Table 1. The authors present the carbon scaffold in descending order, starting from the most abundant. However, have you tried to average (and report) the activity (Ki, IC50, EC50) for each group of cores? would be there any regularity after taking into consideration such approach?

Response: The purpose of Table 1 was to demonstrate the structural diversity of the dataset using carbon scaffold analysis, which is a common practice (ref. https://doi.org/10.1021/acs.jcim.9b00718, https://doi.org/10.1016/j.jhazmat.2020.123724). After following the reviewer's suggestion to attempt to average the activity of each core functional group, no regular conclusions were found.

Q8: Lines 210-213: IIn addition to reporting the number of selected features, it would be beneficial to include their percentage values. For instance, if we originally have 100 features, selecting 50 features would constitute 50% of the total variables.

Response: Thanks to the reviewer's suggestion, we have added the percentage values in line 125-131 marked in red.

Q9: Figure 2: Could you please specify what is plotted on the Y-axis in the graph? Is it the Root Mean Square Error (RMSE) for the model, test, or external test set? Or is it the cross-validation RMSE (RMSEcv)? This detail should be included in the figure caption.

Response: Thank you to the reviewers for their suggestions. The Y-axis in Figure 2 represents the RMSECV, to which we have added a note in the figure caption of Figure2.

Q10: Table 2: How did you estimate the error when reporting Q2, RMSEcv, MAEcv?

Response: The errors of these metrics were determined by calculating the standard deviation for each result.

Q11: Tables 2-4: There are many numbers in those tables. I get the impression that readers who want to analyze/understand the data need another ML algorithm (or at least PCA) to see the trends in this data. Do you think we can do something about that?

Response: We are grateful for the suggestion. These numbers represent the differences between different algorithms and descriptors. Except for the sudden performance changes of individual models, the performance of each algorithm and descriptor is relatively stable. Therefore, we believe that a more intuitive understanding can be obtained through textual descriptions alone, and that the use of algorithms to do additional trend analysis is not very necessary.

Q12: Line 283: Y-randomization is repeated phrase.

Response: Thank you for the reviewer's reminder. The repeated phrases have been removed.

Q13: Model Interpretation: This section lacks a comparison of the authors' results with the general knowledge available in the open literature regarding similar compounds or activities. It should also offer general guidelines on the characteristics of future, highly active molecules, providing direction for researchers involved in synthesizing such compounds. If the authors believe their discussion is sufficient, they should highlight this aspect more prominently in the text.

Response: Thank you very much for pointing this out. We tried to query the predictive modeling of biological activity related to TRPV1 modulators, but the amount of data in only a few reports was too small. The feature structures mentioned in the model interpretation are directions for future highly active molecules.

Q14: RFE-RF: It would be beneficial to include more details about the Recursive Feature Elimination with Random Forest (RFE-RF) method in the Experimental section, as it appears to be a crucial approach utilized in this paper.

Response: RFE-RF has been described in detail in the article, you can read line 115-120 for details.

Q15: Line 306: Please rewrite 'Gini importance of each feature to represent its importance' to 'Gini index for each feature to indicate its significance' (or importance).

Response: Thank you for the reviewer's suggestion. The sentence has been rewritten in the line 232-233 in red.

Q16: Line 322: what exactly you mean by '... featured WORD structures'. Please remove WORD.

Response: Thank you for the reviewer's reminder. ‘WORD’ has been removed from the text (line 271).

Q17: Lines 323-325: I recommend using a color other than gray for the bonds in Figure 9, as the gray bonds are only discernible upon zooming into the substructures.

Response: Thanks to the reviewer's suggestion, the gray bonds in Figure 9 has been replaced with green bonds.

Q18: Figure 9. Please indicate in the caption a meaning of A, B, and C.

Response: Thank you for the reviewer's reminder. The explanations of A, B and C have been added to the caption of Figure 9.

We would like to express our great appreciation to you for comments on our paper. Looking forward to hearing from you.

Yours sincerely,

Tengxin Huang

24 December, 2023

E-mail: h1064482785@163.com

Reviewer 2 Report

Comments and Suggestions for Authors

Some concerns should be addresses as follows:

1. Abstract: future expected investigated based on your theoretical models should be mentioned.

2. Introduction: the aim of this study should be highlighted at the end of the introduction.

3. Methods: the versions of all software used in this study should be mentioned and supported your methods by appropriate references whether they are available. Equations should be numbered

4.   Results and discussion: Fig. 1D-F: PCA: please mention the values of PC1 and PC2, which indicate the probability of the model and explain the variables. Fig. 9 should be improved. The authors should interpret and compare their results with previous studies since the discussion lack suitable references.

5. The authors should scrutinize the manuscript and correct some grammatical mistakes.

Comments on the Quality of English Language

The authors should scrutinize the manuscript and correct some grammatical mistakes. 

Author Response

Dear reviewer:

On behalf of my co-authors, we thank you very much for giving us an opportunity to revise our manuscript, and we also appreciate reviewers very much for their positive and constructive comments and suggestions on our manuscript entitled “Quantitative predictive studies of multiple biological activities of TRPV1 modulators”.

Thank you very much for your comments and professional advice. These opinions help to improve academic rigor of our article. Based on your suggestion and request, we have made corrected modifications on the revised manuscript. We hope that our work can be improved again. The responses to the review comments are as follows:

Q1: Abstract: future expected investigated based on your theoretical models should be mentioned.

Response: The future expected investigation was added to the end of Abstract (line 23-24 marked in red).

Q2: Introduction: the aim of this study should be highlighted at the end of the introduction.

Response: Thanks to the reviewers' suggestions, we have highlighted the aim of the study in the introduction (line 73 marked in red).

Q3: Methods: the versions of all software used in this study should be mentioned and supported your methods by appropriate references whether they are available. Equations should be numbered.

Response: Thanks to the reviewer's suggestion, the references to the software and the numbers of  formulas have been added (line 291, 310 marked in red).

Q4: Results and discussion: Fig. 1D-F: PCA: please mention the values of PC1 and PC2, which indicate the probability of the model and explain the variables. Fig. 9 should be improved. The authors should interpret and compare their results with previous studies since the discussion lack suitable references.

Response: (i) PC1 and PC2 are variables that have been downgraded to be able to represent the molecule more comprehensively in two variables. (ii) Fig. 9 has been improved, and the explanation of Fig. 9 has also been added in the annotation. (iii) The comparisons with previous study results are in the discussion section. (line 273-276 marked in red).

Q5: The authors should scrutinize the manuscript and correct some grammatical mistakes.

Response: We have reviewed the manuscript and corrected grammatical errors.

We would like to express our great appreciation to you for comments on our paper. Looking forward to hearing from you.

Yours sincerely,

Tengxin Huang

24 December, 2023

E-mail: h1064482785@163.com

Round 2

Reviewer 2 Report

Comments and Suggestions for Authors

The authors addressed all previous claims.